# City Scale Traffic Monitoring Using WorldView Satellite Imagery and Deep Learning: A Case Study of Barcelona

Annalisa Sheehan [1,*], Andrew Beddows [1], David C. Green [1,2] and Sean Beevers [1,2]

1   MRC Centre for Environment and Health, Imperial College, London W12 0BZ, UK;
    a.beddows@imperial.ac.uk (A.B.); d.green@imperial.ac.uk (D.C.G.); s.beevers@imperial.ac.uk (S.B.)
2   NIHR HPRU in Environmental Exposures and Health, Imperial College, London W12 0BZ, UK
*   Correspondence: a.sheehan@imperial.ac.uk or asheehan@sgul.ac.uk

**Abstract:** Accurate traffic data is crucial for a range of different applications such as quantifying vehicle emissions, and transportation planning and management. However, the availability of traffic data is geographically fragmented and is rarely held in an accessible form. Therefore, there is an urgent need for a common approach to developing large urban traffic data sets. Utilising satellite data to estimate traffic data offers a cost-effective and standardized alternative to ground-based traffic monitoring. This study used high-resolution satellite imagery (WorldView-2 and 3) and Deep Learning (DL) to identify vehicles, road by road, in Barcelona (2017–2019). The You Only Look Once (YOLOv3) object detection model was trained and model accuracy was investigated via parameters such as training data set specific anchor boxes, network resolution, image colour band composition and input image size. The best performing vehicle detection model configuration had a precision (proportion of positive detections that were correct) of 0.69 and a recall (proportion of objects in the image correctly identified) of 0.79. We demonstrated that high-resolution satellite imagery and object detection models can be utilised to identify vehicles at a city scale. However, the approach highlights challenges relating to identifying vehicles on narrow roads, in shadow, under vegetation, and obstructed by buildings. This is the first time that DL has been used to identify vehicles at a city scale and demonstrates the possibility of applying these methods to cities globally where data are often unavailable.

**Keywords:** deep learning; Earth Observation; high-resolution satellite imagery; object detection; vehicle detection

## 1. Introduction

Accurate traffic data is crucial for a range of different applications such as transportation planning and management, modelling vehicle emissions, and developing spatially detailed exposure data sets for epidemiological studies. Currently, traffic data sets are collected at point locations across cities, using systems such as automatic counters which are mounted in, on, or above the road, or periodic manual counts. These traffic data have limited spatial and temporal coverage, are resource intensive, and the data are often not as widely available as environmental data [1]. Traffic models have also been produced for some cities which are more spatially detailed, however, they do not always cover all roads in a city and rely on detailed, high-quality data inputs [2]. Further, there is no standardised approach for producing transport models, with guidelines on methods, data inputs, scale, and sensitivity analysis varying by country [3]. Conversely, Earth Observation offers a rich, global data resource that is rapidly developing, with data available at an increasingly higher spatial and temporal resolution. Utilising this data to estimate vehicle information is a more cost-effective approach that can be standardised on a global level to provide an alternative to ground-based traffic monitoring in cities [4].

Previous studies have identified vehicles in imagery from the IKONOS and Quickbird satellites, which have spatial resolutions of 0.82 m to 1 m for the Panchromatic (PAN) band

and 3.26 m to 4 m for the Multi-Spectral (MS) band [5], and 0.65 m to 0.73 m for the PAN band and 2.62 m to 2.90 m for the MS band [6], respectively. However, due to vehicles only being a few pixels in size, vehicle extraction using imagery from these satellites has been challenging [4]. With the advent of higher-resolution satellite imagery, such as from the WorldView-3 (WV3) satellite, which has a PAN band spatial resolution of up to 0.3 m, the capability of vehicle extraction has been greatly improved [7,8].

Traditional techniques such as computer vision feature extraction methods and temporal change-detection-based models have previously been used to identify moving vehicles in satellite imagery with varying success. Computer vision feature extraction methods such as scale-invariant feature transform [9], Haar-like features [10] and Histograms of Orientated Gradients (HOG) [11], extract visual features from an image to provide a semantic and robust representation of different objects. However, these methods find it challenging to use a single feature descriptor that extracts all objects in an image due to varying lighting conditions and backgrounds [12]. Temporal change-detection-based methods have had more success in identifying vehicles and extracting vehicle speed and direction from WorldView imagery [4,13–15]. The studies that use temporal change-detection-based methods differ in technical aspects, but follow the same broad structure: firstly, the roads are extracted, secondly, the vehicle location is extracted, and thirdly, it is determined whether a vehicle is moving and if so, the speed is estimated. However, these past studies have focused on a few main roads in small subset areas of cities which tend to be clearly visible in imagery with no objects obstructing the roads and vehicles from view. Therefore, it is unclear whether these methods will struggle when scaled up to identify vehicles in an entire city where the satellite imagery is impacted by shadows, vegetation, narrow roads, and tall buildings.

Over the last decade, there have also been improvements in the ability and accuracy of Deep Learning (DL) Object Detection (OD) algorithms. DL Convolutional Neural Networks (CNNs) are a popular method for OD, due to their ability to identify homogeneous groups of pixels. CNNs are a supervised DL technique that uses a representative training data set to learn patterns and features of objects in an image and to identify the corresponding object classes. DL OD models have commonly been used to detect and classify vehicles and objects in street-level images and traffic videos [16]. More recently, DL OD models have been applied to overhead imagery such as that available from aerial photographs [17–19] and satellites [20–22], to identify objects.

Accurate training sets are fundamental in producing an accurate model, however, they are timely to produce and there is a lack of available training data sets containing vehicles in satellite imagery. The Cars Overhead With Context (COWC) data set consists of vehicles digitised in aerial imagery with a spatial resolution of 15 cm, for six cities in the USA, Canada, New Zealand and Germany [23]. OD models such as RetinaNet [20], YOLT [21] and SIMRDWN [22] have been trained and tested on the COWC data set. Van Etten [21] trained the YOLT model on the COWC data set imagery down-sampled to the spatial resolution of the WorldView-3 satellite (30 cm) and achieved a very high accuracy (F1 score) of $0.9 \pm 0.09$. Other remote sensing training data sets include the xView Challenge data set which contains 30 cm WorldView-3 satellite imagery classified into 60 different object classes including 'small vehicle', 'truck' and 'bus' [24]. The DIOR data set, which consists of optical remote sensing imagery from Google Earth covering 80 different countries with a spatial resolution varying between 0.5 m and 30 m, is classified into 20 object classes including 'vehicle' [25]. In addition to the Dataset for Object Detection in Aerial images (DOTA) which consists of aerial images from a variety of different sensors and platforms with varying spatial resolution and 15 object classes including 'large vehicle' and 'small vehicle' with orientated bounding boxes [26]. These data sets have been used to quantify the accuracy of DL models in detecting objects in remote sensing imagery [27–29]. However, no study has investigated the application of DL OD to satellite imagery to identify vehicles in an entire city. Previous studies have focused on identifying vehicles in overhead imagery and have limited study areas to small subsets of cities (e.g., a few roads).

There are many challenges faced when using satellite imagery for vehicle detection including intrinsic factors of satellite imagery such as off-nadir angle variation, overhead view, infinite orientations, varying illumination, and atmospheric conditions [7,22,30,31]. Remotely sensed images have complex background information, cluttered scenes, and noise interference, making it challenging to detect objects compared to natural, ground-level photos and videos [30]. Additionally, each satellite image covers a large geographical extent and contains a very large number of pixels. For example, a single WorldView satellite image covers over 64 km$^2$ and has over 250 million pixels, resulting in huge file sizes [22,32]. Furthermore, vehicles in satellite imagery are represented by a small number of pixels (approximately 15 to 20 pixels for a car) and therefore remain challenging in high-resolution satellite imagery [20,22,32].

In this study, we trained a DL OD model YOLOv3 on WorldView-2/-3 satellite imagery to identify vehicles across the city of Barcelona. We tested different parameters such as image colour band composition, composition of training data set and model hyperparameters, to understand the impact on the model's accuracy. Additionally, we tested the scalability of these methods to produce a traffic data set for entire cities. The overall aim of this work was to identify vehicle movements in high-resolution satellite imagery to create a city-wide traffic data set that can be used to predict traffic emissions.

## 2. Materials and Methods

### 2.1. Data Description

#### 2.1.1. Study Area

To assess the ability of DL OD to identify vehicles in cities for traffic data set development, the city of Barcelona, Spain, was used as the study area due to its relatively cloud-free skies and being representative of the European vehicle fleet. Barcelona, the capital city of the Catalonia region, is located in the North-East of Spain on the Mediterranean Sea. Barcelona has a population of 1.6 million [33] and an area of approximately 100 km$^2$. The urban fabric of Barcelona varies across the city, from the famous square block building design forming a grid-like pattern in areas such as Eixample, to narrow maze-like streets in the Gothic Quarter and Gràcia.

#### 2.1.2. Imagery

WorldView-2 (WV2) and WorldView-3 (WV3) MS and PAN satellite imagery for Barcelona were used in this study. The spatial resolution of the PAN band is up to 50 cm for WV2 and 30 cm for WV3, and both satellites have a daily sun-synchronous orbit, passing over Barcelona at approximately 11 am. The satellites have eight MS bands including Blue, Green, Red, Coastal Blue, Yellow, Red Edge and two Near-Infrared bands.

In total, ten satellite images were used which had a varying coverage of Barcelona; four images covered the entire city and six images covered smaller subset areas of the city. In this study, for consistency, PAN imagery of 50 cm and MS imagery of 2 m were used for imagery from both satellites. The imagery was pre-processed by Maxar to Level 3, which includes orthorectification with Maxar's finer Digital Elevation Model (DEM) and atmospheric correction. The MS imagery was pansharpened to a spatial resolution of 50 cm using the PCI Geomatics software (version Geomatica 2018) and the University of New Brunswick pansharpening method [34]. The satellite images were split into 1500 × 1500 pixel tiles, which had an overlap of 20 pixels (10 m) to ensure vehicles present at the edges of tiles were not cut off.

### 2.2. Training, Testing and Validation Data Sets

The accuracy of DL OD methods relies on the representativeness of the training data sets. For this study, the training data set consisted of a 5% sample of the ten image scenes, equating to a total area of 34.5 km$^2$. To ensure the training data set was representative, a stratified sampling approach was taken. The imagery was stratified by image day, and 1500 × 1500 pixel tiles were randomly selected from each available day image scene

across Barcelona, this was done to ensure the inclusion of imagery across the year and a wide range of factors inherent to imagery collection such as the off-nadir angle, target azimuth and sun elevation. Additionally, the sample image tiles were checked to ensure good spatial coverage across Barcelona, representing varying topography and different neighbourhood characteristics.

To create the training data set, the sample image tiles were manually digitised in QGIS by drawing a rectangular polygon bounding box (BBOX) around every vehicle present, ensuring a close fit between the BBOX and the vehicle. The vehicles were visually classified into three classes representing modes of movement: parked vehicles, static vehicles, or moving vehicles. The images from the multispectral sensors on the WV2 and WV3 satellites are captured a fraction of a second apart, therefore if a vehicle is moving with a significant speed the vehicle is captured in different positions in the imagery. To differentiate between static and moving vehicles, the position of a vehicle on a road was used and whether a vehicle can be seen to move over multispectral bands. For example, vehicles that are stationary waiting at a traffic light traffic at the time of image capture compared to a moving vehicle in the centre of a road lane which has visible movement over the multispectral images. Examples of the different training data set classes are shown in Figure 1. A total of 19,158 vehicles were digitised, of which, 81% of vehicles were parked cars.

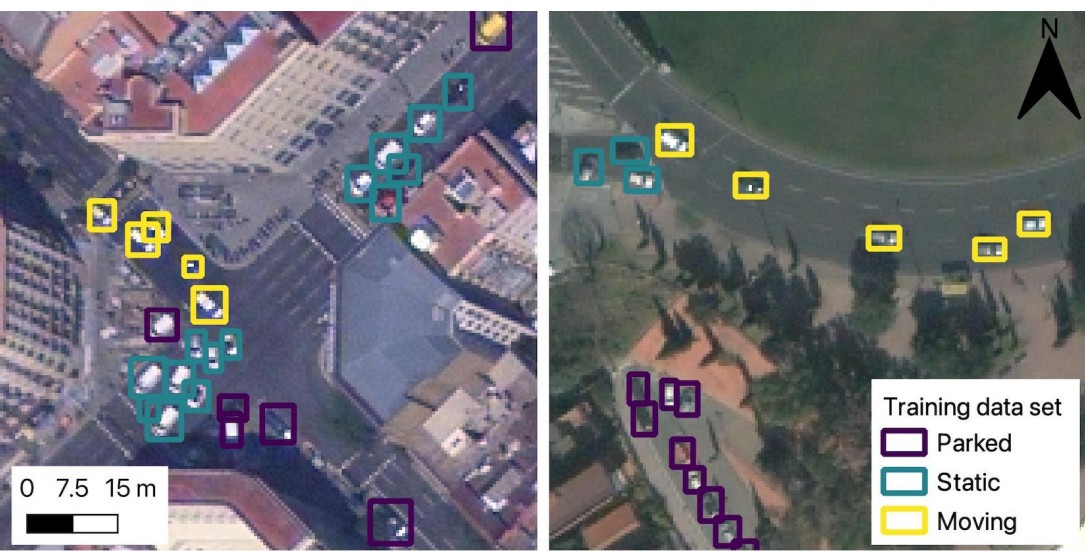

Satellite images © 2019 Maxar Technologies.

**Figure 1.** Images showing the differences between the training data set classes: parked, static, and moving.

The test and validation data sets each included six 1500 × 1500 pixel tiles from six different image scenes and were randomly selected whilst ensuring the images represented different times of year and locations across Barcelona city. A total of 4,148 vehicles were digitised in the test imagery and a total of 2645 vehicles were digitised in the validation imagery.

### 2.3. YOLOv3 and Model Implementation

A supervised DL OD algorithm You Only Look Once version 3 (YOLOv3) by Redmon & Farhadi, [35] was trained to identify vehicles in the satellite imagery. YOLO is a CNN that is built to quickly predict and classify all the objects in an entire image. To do this, YOLO uses a single neural network on an entire image to learn global patterns in the data and incorporate contextual information for all the objects when making predictions for bounding boxes [36]. YOLOv3 makes detections at three different scales by applying a 1 × 1 detection filter on feature maps of three different sizes at three different places in the network using strides of 32, 16 and 8, to downsample the dimensions of the input image [37]. This feature helps the model to identify small objects in imagery. YOLOv3 was

chosen as the model architecture was designed to perform well when identifying small objects, which is key as vehicles in satellite imagery are only tens of pixels in size. The xView-YOLOv3 implementation of YOLOv3 was used [38], as this implementation was designed to run on WorldView imagery.

During the model training phase, the xView-YOLOv3 implementation applied a number of different data augmentation techniques to the full-resolution training data set images and then randomly selected eight $608 \times 608$ pixel-sized chips from the augmented image for training, to improve model performance and avoid model over-fitting. The different data augmentation techniques included: translation (moving image in x or y direction) ($\pm 1\%$), rotation ($\pm 20°$), shear (shifting one part of the image) ($\pm 3°$), scale ($\pm 30\%$), reflection (50% probability, vertical and horizontal), Hue Saturation Value (HSV) scaling: saturation (intensity of colour) ($\pm 50\%$) and value intensity (lightness or darkness of a colour) ($\pm 50\%$).

In total eight YOLOv3 models were trained, each assessing the influence of different training data set compositions, image colour composites and hyper-parameters such as anchor boxes, image size and network resolution, on model performance.

Several different combinations of class labels and training data set compositions were tested. For class labels, models were tested with three classes split by movement type (parked, static or moving) and models with vehicles classified into a single class. Additionally, the composition of the training data set was amalgamated in two different ways: all vehicles (PSM: parked, static and moving), and only static and moving vehicles (SM).

The physical arrangement of the spectral sensors on board the WV2 and WV3 satellites is such that there is a fraction of a second gap between each of the sensors, which can be exploited to identify fast-moving vehicles' speed [4,13,15]. Both Red, Green and Blue (RGB) and Red, Green and Coastal Blue (RGC) composite images were used to train the model and assess the impact on model accuracy as the temporal gap between the Red and Coastal Blue bands (0.29 s) is much longer than the Red and Blue bands (0.015 s), enabling movement of vehicles to be seen.

Anchor boxes are a set of predefined boxes with varying aspect ratios, at multiple scales, and are used by the model during training to help the network to converge and simplify the process of determining the size and location of detection bounding boxes [39]. Anchor boxes specific to the Barcelona training data were created using k-means clustering on the training data set BBOX heights and widths. A total of nine anchors were used as a consequence of the k-means analysis with YOLOv3 using a minimum of three anchor box sizes to make detections at each of the three scales. These were compared to the 30 anchor boxes provided with the xView-YOLOv3 implementation which used the xView challenge data set comprising 60 different object classes in WorldView imagery ranging from passenger vehicles, trucks, fixed-wing aircraft, maritime vessels and buildings [24]. The cluster assignments for the Barcelona training data set is shown in Figure 2. The Barcelona anchor box ratios vary in width and height in pixels between (5.39, 7.85) and (22.96, 28.02) whereas the xView challenge anchor box ratios are much larger, varying from (9.966, 14.84) to (723.9, 517.8) in pixels.

The network resolution parameter can be increased to improve the precision of the network. Through changing the network resolution, an image (of any size) will be upscaled or downscaled to the specific resolution [40]. In theory the larger the network resolution, the more capable YOLO will be at finding smaller objects. Due to the architecture of YOLOv3, the network resolution parameter has to be divisible by 32. For this analysis, the baseline network resolution was 416 pixels, with a network resolution of 608 pixels tested.

During training the YOLOv3 architecture automatically resizes input images to $416 \times 416$ pixel inputs and, therefore, training image tiles larger than the predetermined network image input size may lose a large amount of information during the resizing process. This was an important parameter to investigate as the vehicles in imagery are only tens of pixels in size and thus the network's ability to identify vehicles may be reduced. Two different image sizes were tested: $1500 \times 1500$ pixels and $416 \times 416$ pixels. The

416 × 416 pixel images were created ensuring an overlap of 55 pixels to ensure that no vehicles at the edges of tiles were lost.

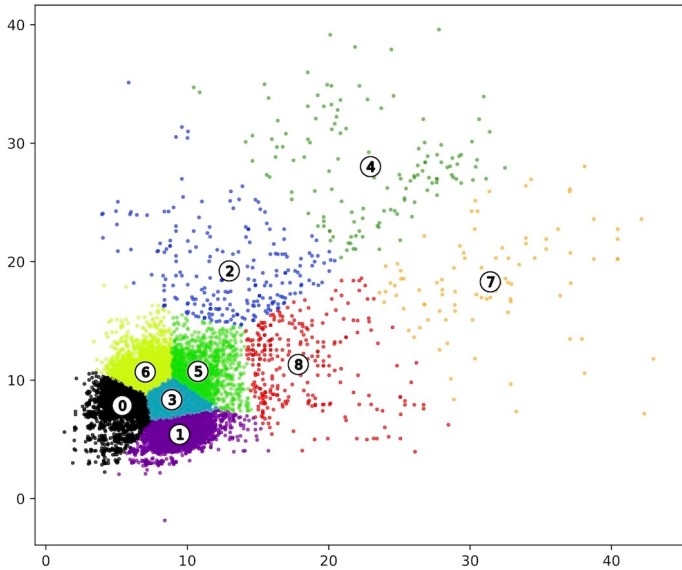

**Figure 2.** K-means cluster assignments and means for Barcelona training data set bounding box widths and heights into nine clusters. The nine different clusters are shown using different colours and labelled 0–8 in the image.

### 2.4. YOLOv8 and Model Implementation

OD is a fast-moving field with new versions of the models almost yearly. As a sensitivity analysis, we compared our results to YOLO version 8n which was released on 10 January 2023 [41]. One of the main differences between YOLOv3 and YOLOv8 is that YOLOv8 does not use anchor boxes, instead, the model predicts the centre of objects [42]. Compared to YOLOv3, YOLOv8n performs better on the COCO data set, obtaining a mAP value of 37.3 compared to 33 for YOLOv3 [35,43]. In contrast to the xViewYOLOv3 implementation used in this paper, the YOLOv8 model has not been optimised for satellite imagery.

### 2.5. Evaluation

The validation data set and different metrics were used to evaluate the performance of the model to identify vehicles including loss function, Intersection over Union (IoU), Precision, Recall and F1 score.

The loss function provides the model with an understanding of predictive performance and enables the model to prioritise certain errors to address during model training. The loss function aggregates the distance scores (errors) for the predictions of the network and the training data into a single numerical value to determine how well the network has done in the example [44]. In YOLOv3 the loss function consists of three parts: localisation loss (centroid x, y, loss and width and height loss), objectness loss (whether a region proposal contains an object, and if so whether the object fits within the training data set classes), and classification loss. The loss value, computed by the loss function during training, can be plotted on a loss plot to visualise the rate of change of a model's learning and show when a model starts to over-train or over-fit. Models start to over-train when the loss has an inflection point and starts to flatten with no change in loss, the loss slowly decreases, or the loss begins increasing.

After model training, the IoU threshold was used to quantify the amount of overlap between the model-predicted BBOXs and the ground-truth BBOXs, and determine whether a model-predicted BBOX is a true positive (Equation (1)). This was done by matching the model detections to the validation data set. If there were multiple model detections for a

ground truth BBOX, the model detection with the highest IoU was selected. Detections with an IoU below 0.5 were removed.

$$\text{IoU} = \frac{\text{Area of Intersection of two boxes}}{\text{Area of Union of two boxes}} \tag{1}$$

range = 0 (no overlap) and 1 (complete overlap).

The number of True Positives (TPs), False Positives (FPs) and False Negatives (FNs) were calculated and the precision (Equation (2)), recall (Equation (3)) and F1 score (Equation (4)) metrics were used to quantify the accuracy of the model detections compared to a ground-truth data set. The precision is the probability of the predicted BBOX matching actual ground-truth BBOXs. The recall metric is the true positive rate and measures the probability of ground-truth objects being correctly detected. The F1 score is a performance metric that provides an overall evaluation of a model by combining the precision and recall metrics into a singular value. This is useful as both the precision and recall metrics assess only one aspect of a model's performance.

$$\text{Precision} = \frac{\text{True Positives}}{(\text{True Positives and False Positives})} \tag{2}$$

range = 0 and 1 (all the model's identified objects are correct).

$$\text{Recall} = \frac{\text{True Positives}}{(\text{True Positives and False Negatives})} \tag{3}$$

range = 0 and 1 (model correctly identified all the objects).

$$\text{F1 score} = 2 \times \frac{\text{Precision} \times \text{Recall}}{\text{Precision} + \text{Recall}} \tag{4}$$

range = 0 (both precision and recall are low: the model did not correctly identify objects and had many false detections) and 1 (both the precision and recall are high: the model identified all objects in an image correctly).

For each model run, two sets of validation were completed: one including a ground-truth data set containing Parked, Static and Moving (PSM) vehicles as a single class, and a second validation with a ground-truth data set containing Static and Moving (SM) vehicles as a single class. The PSM validation was completed as it provided an understanding of how the model performed in identifying all vehicles in the imagery. The SM validation was completed to understand the accuracy of the model to identify moving vehicles, as the overall aim of the work was to quantify traffic flows in a city.

## 3. Results

To identify vehicles in the WorldView imagery, eight different YOLOv3 models and one YOLOv8 model were trained, evaluated, and compared. The different YOLOv3 models targeted different image manipulation processes and fine-tuning of model hyper-parameters, including training class composition, image colour band composition, Barcelona-training data set derived anchor boxes and network resolution.

### 3.1. Loss Plot

To understand the progress of learning over the model training and to limit over-training, the loss plot learning curves were observed. Figure 3a shows the model total loss, precision and recall (averaged every ten epochs) for Model 6 (YOLOv3, RGB, PSM, 1000 epochs, 1500-pixel size image, 9 Barcelona anchors). Overall, the loss plot (Figure 3a) shows that there was a continuous improvement during training, however, over the epochs there was a large variation in total loss, precision and recall. The total loss quickly reduced over the first 250 epochs and fluctuated under a total loss value of 500 for the rest of the training epochs, similarly, the precision and recall rapidly increased over the first

200 epochs before fluctuating between 0.4 and 0.55 for precision and 0.4 and 0.8 for recall, over the remaining epochs.

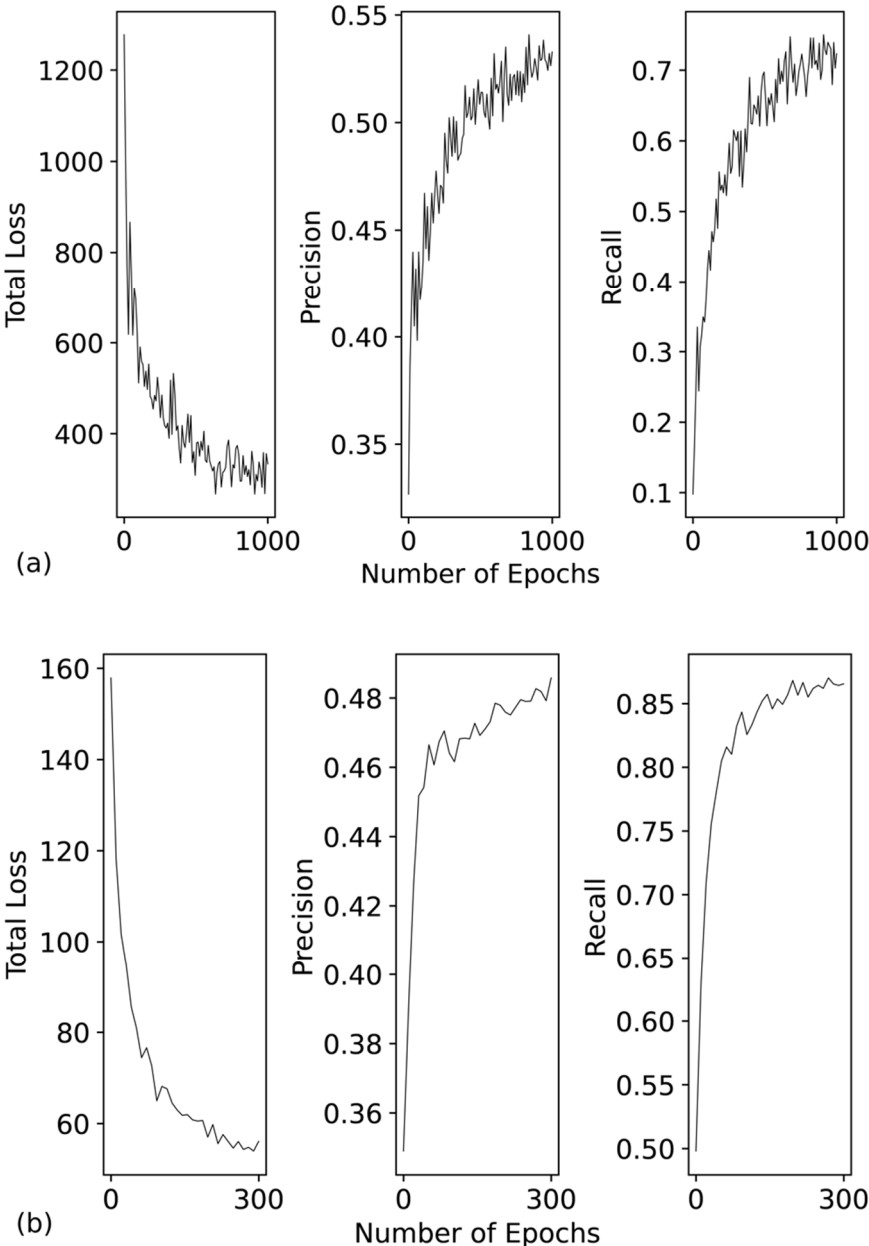

**Figure 3.** Total loss, precision and recall plots averaged every 10 epochs for (**a**) Model 6 (YOLOv3, RGB, PSM, 1000 epochs, 1500-pixel image size, 9 Barcelona anchors) and (**b**) Model 7 (YOLOv3, RGB, PSM, 300 epochs, 416-pixel image size, 9 Barcelona anchors).

The loss plot for Model 7 (YOLOv3, RGB, PSM, 300 epochs, 416-pixel image size, 9 Barcelona anchors), Figure 3b shows that there was less variation in total loss, precision and recall during training compared to Model 6, with the total loss initially rapidly decreasing over the first 75 epochs before steadily declining until 300 epochs. Similarly, precision and recall rapidly improve over the first 75 epochs before fluctuating between 0.45 and 0.5 for precision and 0.8 and 0.9 for recall for the remaining epochs. The larger variation in total loss over training for the 1500-pixel image size model compared to the 416-pixel image size model suggests that the 1500-pixel image size model may struggle with converging. Model convergence is when the model achieves a state where the loss is within an optimal error range and when additional training will not improve the model.

Therefore, the 1500-pixel image size model may benefit from more training examples to help the model converge. The loss plots for the other models are provided in the supporting information (Figures S1–S7).

### 3.2. Confusion Matrix

Confusion matrices are a useful tool to understand classification accuracy as they show the number of objects in each class for both the validation data set (true label) and model detections (predicted label). Figure 4 shows the confusion matrix for Model 1 (YOLOv3, RGB, 3 class, 1000 epochs) where the model was trained with a training data set split into three different classes (Parked, Static and Moving). The fourth class 'Missing' in Figure 4 represents the objects in the validation data set that the model did not detect (FNs). Figure 4 shows that Model 1 correctly classified 55% of objects while 10% of detections were missing a classification and were more likely to classify an object as parked or static compared to moving. The model classification accuracy of 55% suggested that the model is not sufficiently capable of correctly classifying the objects into the three different movement categories. This may be due to the model not having sufficient training examples to learn the difference between parked, static, and moving vehicles, or may suggest that the differences between these categories are too nuanced for a model to distinguish.

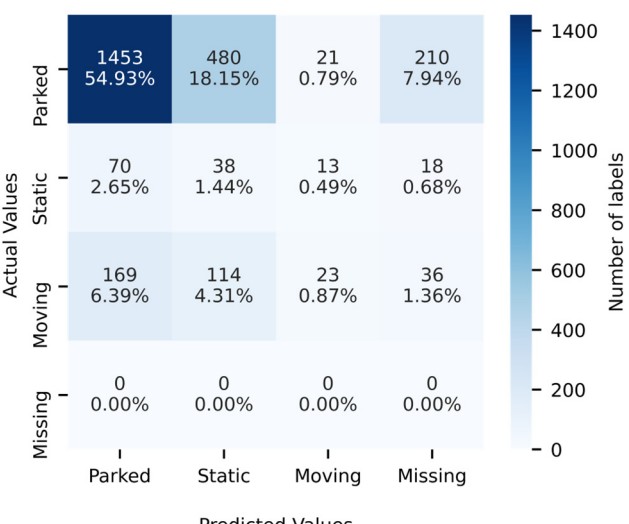

**Figure 4.** Model 1 (YOLOv3, RGB, 3-class, 1000 epochs) class confusion matrix showing the number of predicted detection labels and corresponding true labels. In the model run most detections are accurately classified as parked vehicles.

### 3.3. Performance Metrics of Vehicle Detection

Table 1 shows the validation metrics (precision, recall and F1 score) for the nine models. Both PSM and SM validation for the models were completed to give a deeper understanding of how well the model performs in terms of detecting all vehicles (PSM), and in detecting only static and moving (SM) vehicles as these two modes of movement are important for traffic monitoring. The three top-performing models used YOLOv3 and tested parameters such as Barcelona-training data set derived 9 anchor boxes, input image size (1500-pixel and 416-pixel sized images) and PSM training data set composition. Other parameters such as xView challenge data set 30 anchor boxes, RGC colour composite images, SM training data set and network resolution had a lower precision and therefore a lower overall F1 score. The performance of the model on the test data set is provided in Table S1.

**Table 1.** Vehicle detection model validation statistics. For each model, the Precision, Recall and F1 score accuracy metrics are shown. Two validations were undertaken: PSM and SM where PSM used a validation data set consisting of Parked, Static and Moving vehicles, and SM used a validation data set consisting of Static and Moving vehicles. Model parameters include YOLOv3 or YOLOv8, PSM (Parked, Static and Moving vehicles in a single class), SM (Static and Moving vehicles in a single class) and 3-class PSM (Parked, Static and Moving vehicles split into three separate classes) training data sets, RGB: Red, Green, Blue spectral band image, RGC: Red, Green, Coastal Blue spectral band image, NR: Network Resolution, B-Anchors: Barcelona training data set 9 anchor boxes, XV-anchors: xView challenge data set 30 anchor boxes, Im: Image size in number of pixel width/height, Ep: epochs.

| Model Number | Parameters | Validation Type | Precision | Recall | F1 Score |
|---|---|---|---|---|---|
| 1 | YOLOv3, 3-class PSM, RGB, XV-anchors, Im 1500, Ep 1000 | PSM | 0.50 | 0.86 | 0.63 |
|   |   | SM | 0.09 | 0.88 | 0.17 |
| 2 | YOLOv3, PSM, RGB, XV-anchors, Im 1500, Ep 1000 | PSM | 0.64 | 0.86 | 0.73 |
|   |   | SM | 0.12 | 0.93 | 0.22 |
| 3 | YOLOv3, SM, RGB, XV-anchors, Im 1500, Ep 1000 | PSM | 0.52 | 0.57 | 0.54 |
|   |   | SM | 0.16 | 0.95 | 0.27 |
| 4 | YOLOv3, PSM, RGB, XV-anchors, NR: 608, Im 1500, Ep 1000 | PSM | 0.47 | 0.89 | 0.61 |
|   |   | SM | 0.09 | 0.94 | 0.16 |
| 5 | YOLOv3, PSM, RGC, XV-anchors, Im 1500, Ep 1000 | PSM | 0.56 | 0.87 | 0.68 |
|   |   | SM | 0.11 | 0.94 | 0.19 |
| 6 | YOLOv3, PSM, RGB, B-anchors, Im 1500, Ep 1000 | PSM | 0.69 | 0.79 | 0.74 |
|   |   | SM | 0.13 | 0.83 | 0.23 |
| 7 | YOLOv3, PSM, RGB, B-anchors, Im 416, Ep 300 | PSM | 0.60 | 0.85 | 0.70 |
|   |   | SM | 0.13 | 0.91 | 0.22 |
| 8 | YOLOv3, SM, RGB, B-anchors, Im 1500, Ep 1000 | PSM | 0.63 | 0.19 | 0.29 |
|   |   | SM | 0.47 | 0.78 | 0.59 |
| 9 | YOLOv8n, PSM, RGB, Im 416, Ep 300 | PSM | 0.89 | 0.41 | 0.56 |
|   |   | SM | 0.17 | 0.42 | 0.24 |

In terms of overall accuracy, three models stood out: Model 6, Model 2 and Model 7, with F1 scores of 0.74, 0.73 and 0.70 respectively. Model 6 for the PSM validation achieved a precision of 0.69 and a recall of 0.79. Comparatively, Models 2 and 7 had lower precision values (0.64 and 0.60 respectively) and higher recall values (0.86 and 0.85 respectively). Model 6 differs from Model 2 by using Barcelona-derived anchor boxes, suggesting that Model 6's improvement in precision (5%) was due to the anchor boxes constraining the size of the object the model is looking for, enabling the model to reduce the number of FP detections. Similar patterns were found for other models that used the Barcelona anchors.

Four models had a recall value of 0.93 or above for SM validation: Model 2, Model 3, Model 4, and Model 5. These models tested different parameters: training data set composition (PSM vs. SM), network resolution, and image colour composition (RGB vs. RGC). Interestingly, none of these models used the Barcelona-derived anchor boxes. Of these models, the corresponding SM validation precision values ranged from 0.09 to 0.16, and PSM validation precision values ranged from 0.47 to 0.64. These statistics show that whilst these models were the best performing in terms of identifying the largest proportion of SM vehicles in the imagery, in some cases, the model statistics were badly influenced by the number of FPs. This suggests that the model may not have learnt sufficient contextual information to differentiate a vehicle from other objects and noise in the imagery. However, it should be noted that this may be a limitation of the satellite imagery as it is captured overhead and influenced by factors such as sun elevation and look angle, in addition to the very small number (tens) of pixels representing the vehicle due to the spatial resolution of the imagery (50 cm).

The network resolution parameter was investigated in Model 4 (NR: 608) and can be compared to Model 2 (NR: default 416). Increasing the network resolution parameter is supposed to increase the precision of the network and improve the network's capability of identifying smaller objects. The precision values in Table 1 show that this did not hold true in this case as Model 4 had a 0.17 smaller precision than Model 2 for PSM validation.

Image colour composition (Model 5) had a slight influence on the validation statistics, with false colour RGC images improving the recall by 0.01, however, reducing precision by 0.08 (when compared to Model 2) for PSM validation, with similar results for SM validation. As Model 5 used a training data set with PSM vehicles combined into one category where only 9% of vehicles were moving, these results suggest that using different spectral bands that highlight the movement of vehicles did not improve the models' understanding of what a 'vehicle' is due to the small proportion of moving vehicles.

Two different image sizes were tested: $1500 \times 1500$ pixel images and $416 \times 416$ pixel images (Model 7). From the results, the different pixel size images had a similar overall accuracy shown by the PSM F1 score of 0.70 for Model 7 (416-pixel) and 0.74 for Model 6 (1500-pixel). However, these models varied for the precision and recall metrics, with Model 7 obtaining higher recall values and Model 6 having higher precision values, suggesting that the smaller image size (416-pixel) improved the models' ability to identify vehicles in the imagery (less FNs), whereas the larger image size (1500-pixel) was less prone to erroneously detecting vehicles (less FPs).

The worst performing models in terms of F1 score were both SM models (3 and 8) which had F1 scores of 0.27 and 0.59 for SM validation respectively, and 0.54 and 0.29 for PSM validation respectively. As both these models were trained on just the SM vehicles in the training data set, the poor accuracy suggests that the model did not respond well to the training data set only encompassing a proportion of all vehicles in an image scene. Similarly, the validation results showed that the 1 Class PSM trained model (Model 2), produced higher recall and F1 score values for PSM validation compared to Model 1 (3 Class: Parked, Static, or Moving). Both these cases suggest a lack of contextual difference between parked, static, and moving vehicles for the model to be able to correctly identify either, just the SM vehicles in an image, or classify vehicles into the three categories.

The latest version of YOLO (version 8n) (Model 9: YOLOv8n, PSM, RGB, Im 416, Ep 300) was tested on $416 \times 416$ pixel images and was compared to Model 7 (YOLOv3, PSM, RGB, B-anchors, Im 416, Ep 300). The validation statistics show that overall, the YOLOv3 model (7) performed better with an F1 score of 0.7 compared to 0.56 for Model 9. Model 9 suffered from a high number of FNs, missing nearly 60% of the vehicles in the imagery as shown by the low recall value of 0.41. However, Model 9 had a high precision, (0.89) compared to Model 7's precision value of 0.6. Overall, Model 9 had fewer model detections (1476) compared to Model 7 (4644), which in combination with the validation statistics, suggests that the YOLOv8 model may not have sufficiently learned how to identify a vehicle in the imagery.

To summarise, these statistics have shown that the YOLOv3 model, which has been optimised for satellite imagery, outperformed YOLOv8 using the PSM training data set. Additionally, the YOLOv3 model parameters that had the highest impact on model accuracy included the Barcelona-derived anchor boxes and the smaller 416-pixel input images. The other parameters tended to influence either the number of FPs and the precision values, or the number of FNs and, therefore, the recall value.

### 3.4. Model Detection Maps

To understand these accuracy statistics in more detail and to understand where the models performed well, it is important to look at the objects the models have identified as vehicles. A subset of the validation tiles from Model 6 are shown in Figures 5 and 6. From visually inspecting these figures it was clear that the model (along with most other models) performed well on open, bright, multiple-lane major roads. However, Model 6 found it difficult to identify vehicles in shadow, on narrow streets and obscured by vegetation with

a varying impact on the model's performance. Further, Model 6 (and most of the other models) struggled to identify all vehicles in car parks when the vehicles were parked closely together (Figure 5). Conversely, Model 6 tended to identify vegetation on roads as vehicles and identified non-existent vehicles at the edge of shadows but then missed some vehicles on the edge and in shadows that do exist. However, Model 6 was not impacted to such an extent as other models in identifying parts of building roofs as vehicles.

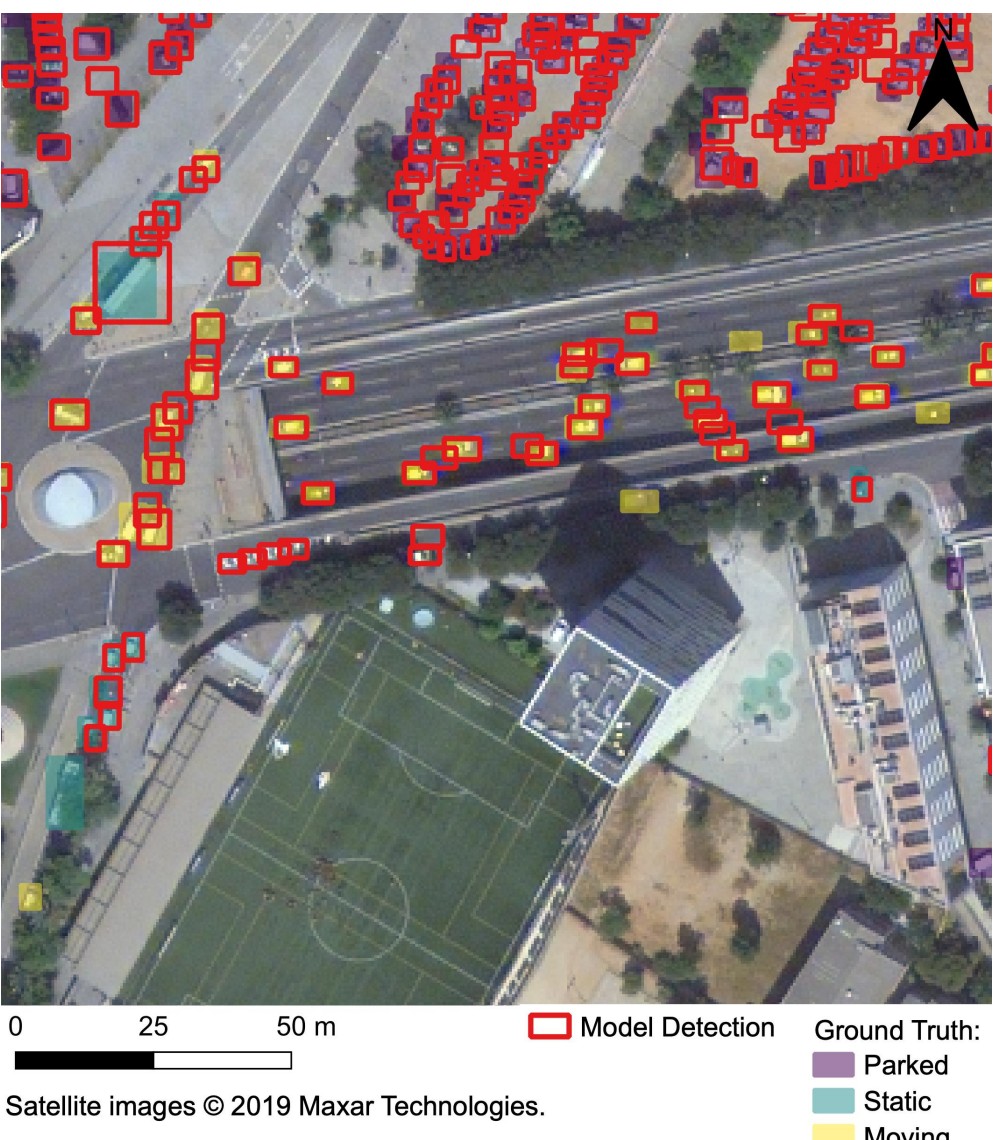

Satellite images © 2019 Maxar Technologies.

**Figure 5.** Satellite image collected in July showing the validation data set (ground-truth) and model detections for an example area of a subset of the model detections for Model 6.

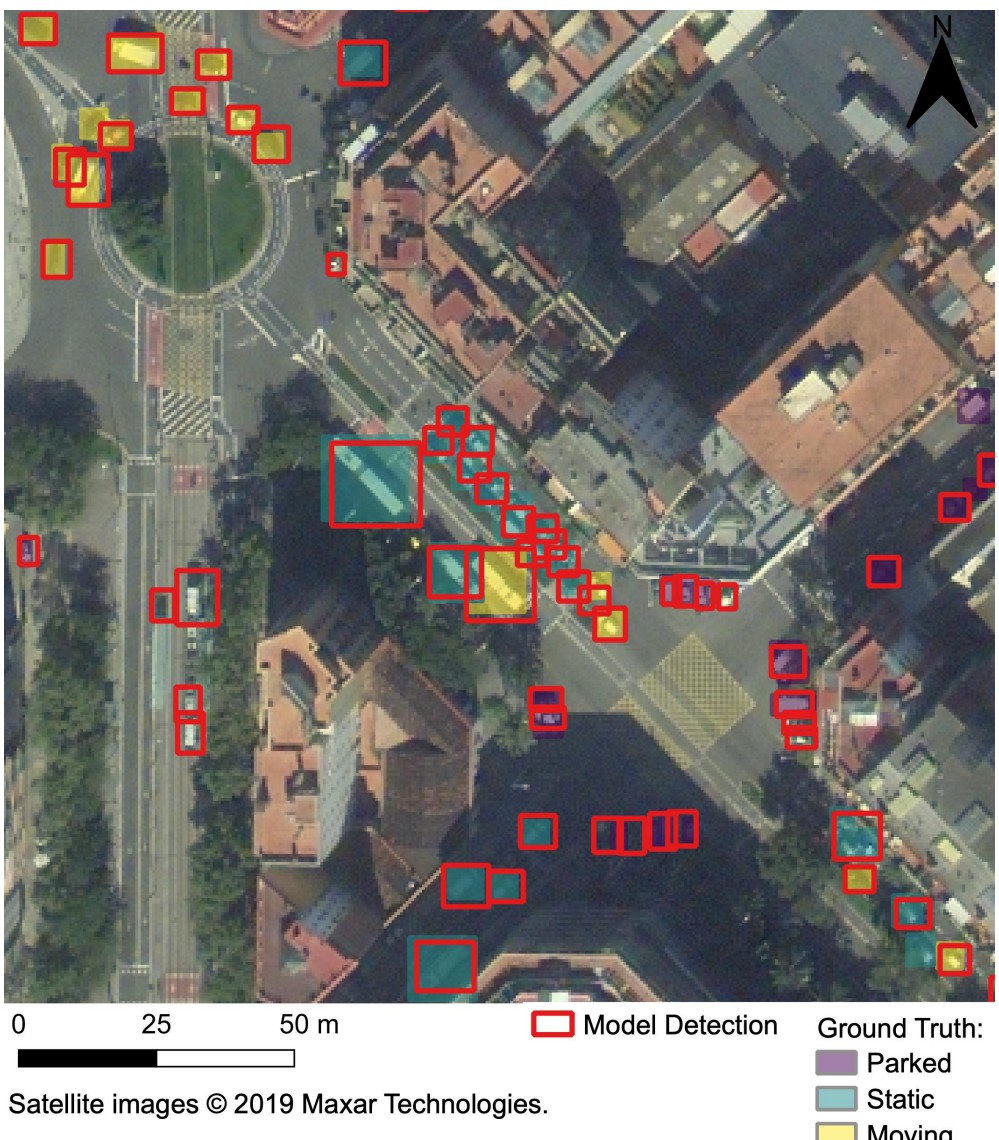

**Figure 6.** Satellite image collected in October showing the validation data set (ground-truth) and model detections for an example area of a subset of the model detections for Model 6.

## 4. Discussion

Overall, eight different YOLOv3 models and one YOLOv8 model were trained to identify vehicles in ten WV2 and WV3 image scenes across Barcelona, collected between 2017 and 2019. The different model runs tested a range of parameters to determine whether they influenced the accuracy of the model, including training data set composition, network resolution, use of anchor boxes, image size and image manipulation. The results showed that the best-performing models included Model 6 (YOLOv3, RGB, 1PSM, 1000 epochs, 1500-pixel image size and 9 Barcelona anchors), Model 2 (YOLOv3, RGB, 1PSM, 1000 epochs and 1500-pixel image size), and Model 7 (YOLOv3, RGB, 1PSM, 300 epochs, 416 image size and 9 Barcelona anchors), achieving F1 scores of 0.74, 0.73, and 0.70 respectively.

Out of all the parameters tested in the Barcelona model, the anchor boxes made the largest difference by reducing the number of FPs and therefore increasing the precision. The validation statistics showed that the models trained with the Barcelona-specific anchor boxes had higher precision and F1 score values compared to the models trained with the xView Challenge anchors for 60 different object classes provided with the YOLOv3 model implementation. This finding agrees with Redmon & Farhadi's study that investigated the impact of handpicked anchor boxes and compared these to running k-means clustering on

the training data set bounding boxes, finding that k-means clustering produced the highest average IoU due to the method's ability to choose representative anchor boxes [45].

The YOLOv3 network resolution sizes of 416 and 608 were compared. This was done because Lin and Chen's study evaluated two different network resolutions (512 and 800), and found that the $800 \times 800$ resolution performed slightly better in terms of F1 score (0.85), however, there was no significant difference between the models [40]. This study suggested that the larger the network resolution the better performing model in terms of F1 score. The results in this study contrast this, as the Barcelona model obtained a higher precision and F1 score using a network resolution of 416 (Model 2) compared to 608 (Model 4), whilst both 416 and 608 had similar recall values, suggesting for this application other fine-tuning techniques and hyper-parameters are more important in determining a model's accuracy. Further work needs to be done to investigate the influence of the network resolution parameter on the 416-pixel imagery training data set.

Different spectral bands were tested to comprise the 3-band input images to the model. The Barcelona results found that using a coastal blue band (Model 5) instead of the blue band (Model 2) resulted in a 0.01 improvement in recall but a 0.08 decrease in precision. The reduction in precision and increase in FPs suggests that through adding the coastal band which shows displacement of moving vehicles, there may not be enough training examples for the model to accurately discern this pattern and is falsely detecting objects and features in the imagery as vehicles. This is consistent with Mace et al.'s study that investigated the use of 8-band spectral imagery in overhead OD and found that the additional bands did not improve model performance [7]. No other studies have investigated the change in accuracy of incorporating different spectral bands into the 3-band (RGB) input images.

Reducing image size was effective at improving the recall metric, however, it resulted in more FPs and to a lesser extent FNs. Van Etten found that by using smaller-sized images, the model was more capable of identifying small objects due to the model resizing the input images to match the network input specifications and, thus, reducing the image resolution [21]. However, models run on 416-pixel images still find it challenging to identify individual cars when they are close together, due to downsampling within the model network. Previous studies have suggested YOLO can find it problematic to identify vehicles where centroids are closer than 32 pixels (16 m) [21,22]. This may explain why the model finds it challenging to accurately identify all parked vehicles in a densely packed environment, resulting in FNs in the models, however, this is less of a problem for SM vehicles.

F1 scores of the models in this study compared comparatively well to other studies that evaluated the accuracy of DL OD models on data sets such as the artwork data set (Picasso) and the Cars Overhead With Context (COWC) aerial imagery data set. YOLOv1 was trained and validated using the Picasso data set and achieved an F1 score of 0.59, whereas R-CNN achieved only 0.226 [36]. The You Only Look Twice (YOLT) model was trained and validated on the COWC aerial imagery data set and achieved an F1 score of $0.9 \pm 0.09$ [21]. A study on detecting small cars in WorldView imagery and comparing different OD models found that the two-stage and multi-stage algorithms such as Faster R-CNN, Grid R-CNN, and Cascade R-CNN achieved maximum F1 scores of 0.75 [27]. Whereas single-stage detectors such as RetinaNet performed slightly worse with an F1 score of 0.719 [27]. The results in this study (Model 6 F1 score: 0.74) are comparable to those found in Groener et al. [27], however, are 16% less than YOLT [21]. This may be due to the higher spatial resolution COWC data set used in YOLT, consisting of aerial imagery that was downgraded to 0.3 m [21] compared to the 0.5 m satellite imagery used in this study. In addition, Van Etten validated the YOLT model on the COWC imagery over Utah which arguably does not have any of the limitations that the satellite imagery over Barcelona has [21]. For example, the Utah aerial images are bright and all directly overhead and have low-rise buildings, and wide streets with little/no impact of shadow, vegetation, or obscured roads.

One of the main factors that impacted the model accuracy was the inherent nature of satellite imagery. Compared to ground-level images, satellites capture overhead imagery and the objects of interest are small, sometimes only a few pixels in size. Satellite images also vary due to factors such as off-nadir angle, target azimuth, sun elevation and scan direction resulting in shadows and roads and vehicles obstructed from view by buildings. Furthermore, the object detection models have been applied to imagery, which contains noisy image scenes, shadows, narrow streets, vegetation, buildings and partially or fully obstructed objects which influences accuracy. However, whilst these factors impact the accuracy of the vehicle detection models, the inherent nature of satellite imagery in this application brings many benefits. As satellites capture imagery across entire cities, vehicles can be identified for the majority of roads in a city apart from the smallest roads (residential roads, living streets and service roads) and, therefore, the resultant vehicle counts have a very high spatial resolution. This contrasts the current point-based systems which can be limited in the number of roads and period of time they can cover due to the cost and reliability of sensors, and the requirement for personnel to deploy sensors or to manually undertake the count in situ or using video imagery. However, it should be noted that whilst the satellite-derived method provides a comprehensive, cost-effective alternative to ground-based traffic data sets in cities, the satellite-derived method is less practical for sparse networks such as rural and motorway roads due to the cost of the imagery.

Finally, there is a lack of training data sets containing vehicles in satellite imagery. This is exacerbated by the range of satellites that collect global high-resolution spectral imagery at varying temporal and spatial resolutions. Therefore, a training data set based on imagery from one satellite may not be applicable to imagery from another satellite, in addition to the wide range of objects in satellite images [22]. To date, there are a limited number of remote sensing data sets with vehicle examples. The data sets that do exist may not include enough examples covering the variety of different cities, environments, and contexts where vehicles are found to sufficiently train a model. For this project, the training data set in Barcelona was manually digitised, taking over 5 months to process 7% of the Barcelona imagery. This is very time-consuming, and due to the relatively small size of the objects, is also a challenging task. To progress in this field of research, it would be beneficial to create a central repository of training data sets, however, this will need to be done in collaboration with the satellite imagery providers.

## 5. Conclusions

This study was the first attempt at utilising very high-resolution satellite imagery and a Deep Learning Object Detection model (YOLOv3) to identify vehicles, road by road, in an entire city. The results showed that the best performing model (Model 6) had a 0.74 overall accuracy in identifying vehicles and performed comparatively well compared to previous studies using DL OD methods. This work showed that satellite imagery offers an approach that could generate a global data set, enabling vehicle count to be estimated road by road across entire cities, with up to a daily resolution from an archive of imagery. This is important, as global traffic data sets are essential in a range of different applications such as transportation planning and management, and in establishing the environmental health impact of vehicle air pollution and greenhouse gas emissions. The global potential of this method is especially important for low- and middle-income countries (LMICs) which have some of the greatest environmental health problems, with the greatest city population growth and limited resources.

**Supplementary Materials:** The following supporting information can be downloaded at: https://www.mdpi.com/article/10.3390/rs15245709/s1, Figure S1: Total loss, precision and recall plots averaged every 10 epochs for Model 1: YOLOv3, 3class PSM, RGB, XV-anchors, Im 1500, Ep 1000; Figure S2: Total loss, precision and recall plots averaged every 10 epochs for Model 2: YOLOv3, PSM, RGB, XV-anchors, Im 1500, Ep 1000; Figure S3: Total loss, precision and recall plots averaged every 10 epochs for Model 3: YOLOv3, SM, RGB, XV-anchors, Im 1500, Ep 1000; Figure S4: Total loss, precision and recall plots averaged every 10 epochs for Model 4: YOLOv3, PSM, RGB, XV-anchors, NR: 608, Im

1500, Ep 1000; Figure S5: Total loss, precision and recall plots averaged every 10 epochs for Model 5: YOLOv3, PSM, RGC, XV-anchors, Im 1500, Ep 1000; Figure S6: Total loss, precision and recall plots averaged every 10 epochs for Model 8: YOLOv3, SM, RGB, B-anchors, Im 1500, Ep 1000; Figure S7: Training data set box loss, precision and recall plots averaged every 10 epochs for Model 9: YOLOv8n, PSM, RGB, Im 416, Ep 300; Table S1: Vehicle detection YOLOv3 models validation statistics using the test data set. For each model the Precision, Recall and F1 score accuracy metrics are shown. Two validations were undertaken: PSM and SM where PSM used a validation data set consisting of Parked, Static and Moving vehicles, and SM used a validation data set consisting of Static and Moving vehicles. Model parameters include: PSM (Parked, Static and Moving vehicles in a single class), SM (Static and Moving vehicles in a single class) and 3 class PSM (Parked, Static and Moving vehicle split into three separate classes) training data sets, RGB: Red, Green, Blue spectral band image, RGC: Red, Green, Coastal Blue spectral band image, NR: Network Resolution, B-Anchors: Barcelona training data set 9 anchor boxes, XV-anchors: xView challenge data set 30 anchor boxes, Im: Image size in number of pixel width/height, Ep: epochs.

**Author Contributions:** Conceptualization, S.B.; Data curation, A.S.; Formal analysis, A.S.; Funding acquisition, S.B.; Methodology, A.S., A.B., D.C.G. and S.B.; Supervision, A.B., D.C.G. and S.B.; Validation, A.S.; Visualization, A.S.; Writing—original draft, A.S.; Writing—review & editing, A.S., A.B., D.C.G. and S.B. All authors have read and agreed to the published version of the manuscript.

**Funding:** This research was funded by the Natural Environment Research Council via the London NERC Doctoral Training Partnership, grant number NE/L002485/1.

**Data Availability Statement:** The YOLOv3 trained weights for Model 6 are deposited in Zenodo: https://doi.org/10.5281/zenodo.8017891 (accessed: 9 August 2023).

**Acknowledgments:** The authors thank the NERC Earth Observation Data Acquisition and Analysis Service (NEODAAS) for access to compute resources for this study. This study is partly funded by the National Institute for Health Research (NIHR) Health Protection Research Unit in Environmental Exposures and Health, a partnership between the UK Health Security Agency and Imperial College London. The views expressed are those of the authors and not necessarily those of the NIHR, UK Health Security Agency or the Department of Health and Social Care. Imagery made available as part of the European Space Agency project on Satellite Air Quality Modelling.

**Conflicts of Interest:** The authors declare no conflict of interest.

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
