# Peer review of "City Scale Traffic Monitoring Using WorldView Satellite Imagery and Deep Learning: A Case Study of Barcelona"

_remotesensing, doi:10.3390/rs15245709_

Round 1

Reviewer 1 Report

Comments and Suggestions for Authors

Summary:

Accurate traffic data is crucial in many applications like transportation planning. Using advanced deep learning based object detection methods on high resolution satellite imagery has great potential in city level vehicle identification and traffic counting. 

The authors used YOLOv3, one of the state-of-art deep learning models in object detection, on WordView-2/3 satellite imagery (0.5m resolution) on vehicle detection tasks on city level (in Barcelona, Spain). Specifically, the authors collected WordView-2/3 multispectral (MS) and panchromatic (PAN) bands data pre-processed to Level 3, and pansharpened them into 0.5m resolution, split into 1500 x 1500 pixel tiles. They selected 5% tiles and manually labeled vehicles for training data, and used a separate set of 6 tiles as validation data. They conducted training using xView-YOLOv3 model implementation with several different hyper-parameters, and evaluated with metrics like loss function, precision, recall, F1-score on validation data. The results were compared among different hyper-parameters.

The overall writing is clear and easy to follow, experiments are solidly designed, and has innovative contributions as it’s the first work on city level car detection using YoLoV3 and WorldView data. However, there are some missing or confusing descriptions, the content is not very comprehensive. Besides, their model and data choices need reconsideration or better explanations (see details in comments below).

General comments:

1.   The authors used YOLOv3, which is a state of art model for object detection. However it’s not best fit for small objects. As a comparison, YOLT (You Only Look Twice) was proposed in 2018 as an extension of YOLO and is specially designed for small object detection on remote sensing imagery. The model is open sourced and there are also newer versions. The authors did mention YOLT had a good performance in a related work, though on a different data set (line 461). All of these make YOLT seem a better alternative and is worth trying for comparison with YOLOv3.

2.  The authors used both WorldView-2 and WorldView-3 data. But WV2 has a lower resolution (~0.5m) than WV3(~0.3m), and therefore the eventual resolution they used is 0.5m. It’s unclear why they mixed WV2 and WV3 here. If they just used all WV3 data and the eventual resolution could have higher resolution (~0.3m), would that result in higher accuracy? It also helps to see how the spatial resolution affects the detection accuracy here.

3. The authors did the result evaluation only on validation data set, i.e. only 6 750mx750m tiles. This to me is OK but the estimated accuracy result is less convincing. Ideally the authors should have a separate test data set which have larger spatial coverage, and not visible during hyper-parameter tuning (ie. Table-1). The overall accuracy should be estimated and reported on the test data set. An alternative way is to cross-validate their result with an external sourced data set (if that exists).

Specific comments:

1.  In line #151-152, it’s mentioned that labels were classified into 3 vehicle types, including passenger car, LDV, HDV. But these types never appeared later. It’s unclear if these types were used or not in later experiments. 

2. Can you give more details about parked vs static cars? How do you clarify their differences, especially how you tell them apart from the imagery when labeling vehicles?

3. In table1, what’s the difference between 3 class PSM and PSM training data set? Is that PSM considers all Parked/Static/Moving as a single class (so the object detection is single-class detection task), and 3 class PSM considers Parked/Static/Moving as separate classes (so the objection detection is 3-classes detection task)? And how is validation done? For the PSM validation set, Is that done by treating all PSM cars as single-class? And for the SM validation set, is that done by treating all SM cars as a single-class? Please clarify these in paper.

4.  What’s the IoU threshold value you used to define the match between predicted and truth bounding box? Please clarify this in paper.

5.  In line #176, it’s mentioned the data augmentation is done for 608 x 608 pixel tiles. However, in line #208, it’s mentioned that the YOLOv3 model takes input of 416 x 416 pixels. Should these 2 resolutions be matched, and be the NR (network resolution) you mentioned in table1? In other words, when NR is default 416 pixels, the data augmentation is done for 416 x 416 pixels and input has 416 x 416 pixels; and when when is 608 pixels,  the data augmentation is done for 608 x 608 pixels and input has 608 x 608 pixels, is that right? Please correct or clarify in paper. 

6. In line #415-417, it’s mentioned “The validation statistics showed that the models trained with the Barcelona specific anchor boxes outperformed the models trained with the xView Challenge anchors..”. However, looking at table 1, when comparing model 2 and model 7, model 2 (with xView anchors) has higher recall and model 7 (with B-anchors) has higher precision. Similar results can be found between model 3 and 8. Therefore, it doesn’t look correct to say that B-anchor outperformed xView anchors as both recall and precision should be considered. 

7. In line #492-495, it’s mentioned that “satellite derived method is less practical for sparse networks such as rural and motorway roads due to the cost of the imagery”. Can you explain why satellite imagery has a higher cost on a sparse network?

8. To explain why anchor box hyperparameters (B-anchor vs xView-achor) affects the accuracy, the authors should give more details on the anchors, like their clustering results, and the sizes and ratios of the anchor boxes, and how the B-anchor boxes differ from xView anchor boxes.

9. Since the Figure 2 shows a trend of misidentifying static and moving vehicles to parked vehicles in Model 1, have you considered techniques to rebalance your training classes? I noticed your training samples have 81% parked vehicles in line 155.

Reviewer 2 Report

Comments and Suggestions for Authors

The paper presents a deep-learning approach for vehicle detection across the urban expanse of Barcelona. The authors display commendable dedication in working with an extensive city-scale dataset, and the writing effort is notably apparent. However, the paper would benefit from an enhanced organizational structure and the introduction of more cutting-edge and inventive contributions.

Several concerns warrant consideration:

-The majority of references appear outdated.

-Comparative analyses with other methods are absent.

-The lack of dataset examples is apparent.

-In line 159, it would be valuable to incorporate sample images representing diverse car types within the datasets.

-The choice of the YOLO version for detection, YOLOv3, could be questioned for not utilizing a more recent version.

- There are no statistics from using times series images or city scale level only the total number of cars.

-On line 189, providing visual samples illustrating parked, static, and moving vehicles would enhance clarity.

-The rationale behind training with dimensions of 1500 x 1500 (line 213) ought to be explained, as its feasibility might be questionable.

-Figure 1's utilization of differing epoch numbers in the two training groups requires clarification.

-A rationale for not extending the number of training epochs until graphical stability is achieved should be provided. Additionally, the omission of models 1-5 in Figure 1 needs addressing.

-Line 297 should delineate the distinction between parked and static vehicles, along with the model's means of differentiation. Given their visual similarity, examples of each category are essential. The differentiation of moving vehicles could also be clarified through visual aids.

-Figures 3 and 4 need to depict various categories present in images, including ground truth rectangles for concealed vehicles.

-The discussion section contains elements better suited for the related work section.

Reviewer 3 Report

Comments and Suggestions for Authors

We know that object detection is a relatively mature research direction with a large number of methods. The YOLO series has developed to YOLOV8, but this article only uses YOLO v3, which is a relatively outdated method. Moreover, the author has not made any modifications to the YOLO v3 model, and there is a lack of innovation in the algorithm.

In the experimental section, the author only compared the training results under different data input conditions. It is recommended that the author compare the latest object detection models, including transformer based object detection.

The data used by the author is relatively single and the amount of data is not large. The training of deep learning generally requires a large number of samples, and it is recommended that the author increase the number of training samples.

Reviewer 4 Report

Comments and Suggestions for Authors

This paper used high-resolution satellite imagery and deep learning methods to identify vehicles at a city scale. Honestly speaking, the constructed dataset for vehicle detection was valuable. However, the applied YOLOv3 method and the experimental analysis lack innovation and novelty.

(1) In the abstract and introduction, the paper described that “this is the first time that DL has been used to identify vehicles”. There are lots of studies on vehicle detection using remote sensing images. Some datasets for detecting vehicles have already been constructed, such as DIOR (Object Detection in Optical Remote Sensing Images: A Survey and A New Benchmark), and DOTA (DOTA: A Large-scale Dataset for Object Detection in Aerial Images). These datasets contain different kinds of vehicles, which can be used to train the detection model.

(2) The authors classified the vehicles into three modes of movement: parked vehicles, static vehicles (vehicles that are stationary in traffic at the time of image capture e.g. waiting at a traffic light), or moving vehicles. I am confused with these three modes. The author should provide some examples or images to show the differences between these modes.

(3) In the Results Section, the loss plot and confusion matrix were demonstrated. On the one hand, there is no need to show these results as it is the basic results of the deep learning method. On the other hand, in Figure 1, the total loss did not converge. Thus, the precision and recall in Figure 1 can be further improved when increasing the training epochs.

(4) As I suggested in comment 3, the number of epochs was too small to obtain robust results. As a result, the comparison results in Table 1 were not reliable, and cannot be analyzed to verify the effectiveness of different models.

(5) In section 3.4 Model detection maps, the ground truth should be displayed using the transparent box. Besides, the model detection should contain three colours to show three kinds of vehicles: parked, static, and moving.

(6) This paper applied YOLOv3 to detect vehicles. Why not use other existing latest approaches?

Round 2

Reviewer 4 Report

Comments and Suggestions for Authors

No more comments. The proposed issues have been revised.